# A Narrative Hypothesis: The Important Role of Gut Microbiota in the Modulation of Effort Tolerance in Endurance Athletes

**DOI:** 10.3390/nu17172836

**Published:** 2025-08-31

**Authors:** Jesus Álvarez-Herms, Martin Burtscher, Francisco Corbi, Adriana González, Adrián Odriozola

**Affiliations:** 1Hologenomiks Research Group, Department of Genetics, Physical Anthropology and Animal Physiology, University of the Basque Country UPV/EHU, 48080 Leioa, Spain; jesusah80@gmail.com (J.Á.-H.); adriana.gonzalez@ehu.eus (A.G.); 2Phymolab, Physiology and Molecular Laboratory, 40170 Collado Hermoso, Spain; 3Department of Sport Science, University Innsbruck, 6020 Innsbruck, Austria; martin.burtscher@uibk.ac.at; 4Institut Nacional d’Educació Física de Catalunya (INEFC), 25192 Lérida, Spain; f@corbi.neoma.org

**Keywords:** microbiota, exercise, pain, effort perception, nutrition, endurance exercise

## Abstract

**Background:** Regulating sensations of fatigue and discomfort while performing maximal endurance exercise becomes essential for making informed decisions about persistence and/or failure during intense exercise. Athletes with a higher effort capacity have competitive advantages over those with a lower one. The microbiota–brain axis is a considered the sixth sense and a modulator of the host’s emotional stability and physical well-being. **Objectives:** This narrative review aims to explore and evaluate the potential mechanisms involved in regulating perceptions during endurance exercise, with a focus on the possible relationship between the gut microbiota balance and the neural system as an adaptive response to high fatigue chronic exposure. **Methods:** Electronic databases (PubMed, Web of Science, Google Scholar, and Scopus) were used to identify studies and hypotheses that had documented predefined search terms related to endurance exercise, gut microbiota, the central nervous system, pain, discomfort, fatigue, and tolerance to effort. **Results:** This narrative review shifts the focus concerning the symbiotic relationship between the gut microbiota, the vagus nerve, the central/enteric nervous system, and the regulation of afferences from different organs and systems to manage discomfort and fatigue perceptions during maximal physical effort. Consequently, the chronicity supporting fatigued exercise and nutritional stimuli could specifically adapt the microbiota–brain connection through chronic efferences and afferences. The present hypothesis could represent a new focus to be considered, analysing individual differences in tolerating fatigue and discomfort in athletes supporting conditions of intense endurance exercise. **Conclusions:** A growing body of evidence suggests that the gut microbiota has rapid adaptations to afferences from the brain axis, with a possible relationship to the management of fatigue, pain, and discomfort. Therefore, the host–microbiota relationship could determine predisposition to endurance performance by increasing thresholds of sensitive afferences perceived and tolerated. A richer and more diverse GM of athletes in comparison with sedentary subjects can improve the bacteria-producing metabolites connected to brain activity related with fatigue. The increase in fatigue thresholds directly improves exercise performance, and the gut–brain axis may contribute through the equilibrium of metabolites produced for the microbiota.

## 1. Introduction

In recent years, research on the connection between the gut microbiota (GM) and the central nervous system (CNS) has been growing exponentially [1,2], particularly in its relation to health, physical performance, and cognitive processes related to perception, emotions, and pain [3,4,5]. Until now, only a limited number of studies have reported the interaction between the GM and the brain’s emotions, perceptions, and sensations. Thus, no previous studies have hypothesised on the possible bidirectional influence existing between endurance exercise and the GM-CNS axis, which modulates fatigue and unpleasant sensations [1,6]. The primary communication networks between the gut and brain involve neural, endocrine, and immune pathways, encompassing the central, enteric, and autonomic nervous systems, as well as the hypothalamic–pituitary–adrenal axis. The GM can modulate the enteric nervous system response through the excitability of the nervous system, as well as the production of metabolites that enable signals to reach the CNS via the vagus nerve [7]. The GM, composed of a complex community of microorganisms, interact with the vagus nerve in the passage of digestive material [8] and in the regulation of gut hormones, neurotransmitters, and short-chain fatty acids (SCFAs) [8,9,10]. Therefore, individual differences and chronic changes in the GM profile may influence the cognition, perceptions, and sensations of the host. Some investigations have described that exercise in humans also increases neurogenesis and alters the microbiota profile [11].

Endurance exercise involves the ability to sustain a specific type of physical activity, such as running, cycling, or swimming, for a prolonged period, where intensity and duration increase, perceived effort rises, and fatigue levels become limiting to continue the activity. The physiological changes that occur during long endurance activities, affecting different systems such as cardiovascular and respiratory, metabolism, thermoregulation, and acid–base balance, contribute to altered perceptions. For this reason, training routines contribute to elevating sensitisation, the neuronal plasticity that occurs in response to prolonged fatigue, pain, and discomfort stimuli during intense efforts, even over several days [8]. Therefore, the globality of training stimulus, supported by endurance athletes, focuses on improving, on the one hand, physiological performance, and, on the other, enhancing sensitivity to tolerating discomfort and unpleasant sensations.

The hardness perceived during endurance exercise may contribute to increased sensitisation for peripheral tissues, as well as GM adaptation in athletes. The present hypothesis describes how promoting a higher effort sensitisation from peripheral and central tissues adaptively changes the abundance of specific gut bacteria related to brain perceptions (see Figure 1). Overall, in response to physical and cognitive stress, two different regulatory systems are activated: (1) the sympathetic–adrenomedullary and hypothalamic–pituitary–adrenal axes [8,12], and (2) the autonomic nervous system (ANS) [13]. Throughout these processes, the intestine is particularly important due to its direct influence on the regulation of the CNS, among other factors. Not surprisingly, there is increasing interest in research on GM and the brain axis, as well as their various connections with health, physical performance, and cognitive processes related to perception, emotions, and pain [3,4,5]. Recently, several publications have reported on the implications of GM on physiological and cognitive functions [1,6]. Still, data are largely lacking on how gut health can directly influence endurance performance through regulated emotions and tolerance to discomfort and fatigue.

Notably, it has been suggested that interventions enhancing GM diversity could positively impact endurance performance [14] and, in parallel, stimulate the production of metabolites such as NTs and other molecules involved in modulating perceptions and, consequently, tolerance to fatigue [15,16,17]. Instead, a GM imbalance may lead to poor well-being from leaky gut syndrome, which includes chronic low-grade inflammatory conditions, emotional instability, dysregulation of neurotransmitters, chronic fatigue, and even depression, all of which may impair athletic performance and tolerance to fatigue [18,19].

In this article, we reviewed the current scientific evidence on the relationship between endurance exercise and GM modulation in CNS perceptions. In addition, we discussed whether there exists a possible GM profile that improves higher effort tolerance. The main goal of the present article was to explore the hypothesis based on how hardness training in athletes could influence the GM ecosystem, alter sensory control through the autonomic nervous system, and influence additional gut metabolites in the blood and achieving the brain. The current work aimed to broaden perspectives in the field of sports science by suggesting that GM composition plays an important role in modulating perceptions during endurance performance.

## 2. The Relationship Between Endurance Exercise, Fatigue Perceptions, and Gut Microbiota Adaptations

The factors limiting endurance performance have traditionally focused on understanding how cardiovascular, metabolic, and muscular adaptations are produced [20]. The central (mental) regulation and the control of peripheral afferences from specific locomotor and visceral tissues that limit endurance performance through perceptions remain less studied. Human feeling includes the expression of body perceptions, emotions, and the modulation of systemic afferences. In the complex interplay of all these influences, the gastrointestinal tract and the GM, particularly, are relevant components influencing higher cerebral functions and behaviour [21,22,23]. In this regard, the gut metabolomic dialogue with the brain is essential for regulating perceptions and sensations [24], acting as both an autocrine, paracrine, and endocrine organ. In addition, the relationship between endurance exercise and this malleable organ, composed of the GM, could be very important to modulate physiological perceptions related to physical effort tolerance.

In such a case, discomfort and fatigue represent an individual and subjective perception involving nociception and nociplasticity [25,26]. In some cases, a higher capacity for tolerating discomfort during endurance exercise or other activities can explain why some individuals achieve better physical performances than others [27,28,29]. Hence, pain and discomfort coping is an integral part of athletic preparation that develops athletic character [30,31,32,33,34,35,36,37,38] in endurance athletes [33,39,40,41,42,43,44,45,46,47] or combat fighters [48], compared to power athletes [49].

The “individual limit of sensory tolerance” [28,29,50,51,52] has been proposed to describe the point at which individuals recognise their limit based on fatigue, pain, or effort [53], and can be improved following several weeks of intense training [44,54]. Pain, defined as the unpleasant, noxious perception that affects mood, social life, and overall quality of life, is a significant limiter of physical activity and quality of life. The individual threshold of pain is subjective, involving not only nociception but also emotional, cognitive, and social components [25]. During maximal endurance activities, athletes feel unpleasant sensations described with comments such as the following: “I cannot continue the race at this pace”, “exercise was hard”, “I never feel good”, “legs do not work”, etc. This effort fatigue is initiated by the activation of nociceptors that populate peripheral organs, such as skin, muscles, bones, joints, and deep visceral tissues [25,55]. Acute pain can contribute to modulating immune responses and protect organisms; however, chronic and sensitive thresholds in the nervous system can lead to elevated nociceptive activation, making organisms more sensitive to minimal immune reactions. Previous studies in clinical medicine have linked the composition of the gut microbiota to a higher incidence and progression of hyperalgesia and pain in certain chronic pathologies and treatments, such as chemotherapy [56].

Adaptive mechanisms for increasing pain and discomfort tolerance primarily depend on the stimulation of the interoceptive system, which is interconnected with the brain and integrated into the interoceptive consciousness [57,58,59]. The interoceptive system is an essential peripheral governor, constantly sensing physiological changes that trigger homeostatic thresholds for internal perceptions, emotions, pain, temperature, oxygen, sensory touch, muscle tension, discomfort, and intestinal sensations. These sensations are processed in the brain as an integrated interoceptive conscience [57,58,59]. As depicted in Figure 2, physiological afferences processed by the brain from peripheral organs and tissues transfer a consciousness of the internal body state during physical activities [60]. Thus, the operation of the interoceptive system is a highly relevant topic of research interest for manipulating perceptions and cognitive responses in conditions of effort, pain, or discomfort [61].

The intensity and duration of endurance exercise imply metabolic alterations that modify the neural response in the motor cortex, resulting in an increase in exertion from a perceptual perspective of sensations [62,63]. Previous studies have suggested that GM can modulate the neural system originating from the intestines and may influence pain thresholds in the brain, which are related to various systemic conditions [3,64,65]. The GM is capable of activating nociceptors [66], altering mediators of inflammation, and inducing phosphorylation in certain receptors and ion channels of sensory neurons, which can result in peripheral sensitisation [65]. Under this premise, the chronic stimulus of endurance exercise and other specific routines, such as nutrition and hydration, changes the GM profile individually. Therefore, the individual sensitive perceptions in a complex system of the body include the afferences derived from the gut through neural, chemical, and molecular signals.

## 3. The Hypothesis: Connection Between the GM and Tolerance to Effort, Pain, or Discomfort Sensations

During maximal endurance exercise, the physiological demands increase proportionally. The brain acts mainly in two protective ways: (1) monitoring biological vital signals from specialised cells for oxygen, temperature, vascular pressure, etc., and (2) producing neural perceptions according to the individual thresholds (set-points) that are derived from afferent discharges from organs and tissues to optimize the intensity of exercise. In addition, gut–brain communication is crucial in regulating afferent responses related to all stimuli arising from immune responses, inflammatory pathways, and metabolic efficiency [67], as well as systemic organs (see Figure 2). With regular endurance training, the different systemic thresholds can alter their grade of tolerance as an adaptive vital mechanism; if it is positive, athletes can perform at higher exercise intensities because their physiological and perceptual limits are increased based on their brain perception tolerance.

Concomitant changes that occur during exercise alter oxygenation, temperature, and metabolic activity, which directly affect gut microenvironments and thereby impact the gut microbiota [68]. The important role of the GM during physical failure is considered due to its modulatory effect on metabolic digestion, the immune system, systemic inflammation, and endocrine function, which involves regulating NTs and hormones related to perceptions, emotions, and sensations [1,6,69]. Here, we hypothesise that tolerance to pain, fatigue, and discomfort may be linked, in part, to intestinal barrier health, damage, and its biological function of permeability. This premise is based on the concept that the gut microbiota can directly or indirectly modulate peripheral sensitisation underlying chronic afferent pain or unpleasant sensations through multiple mediators, including the microbial by-products (e.g., PAMPs), metabolites (e.g., SCFAs, Bile Acids), and neurotransmitters (e.g., GABA) that are released. In this regard, athletes who are more sensitive to pain and effort sensations may be more susceptible to dysregulation of neurotransmitters and may experience GM dysbiosis [70]. In this case, in response to exercise demands, the GM may be unable to produce and maintain a certain threshold of SCFAs and NTs in the brain [70,71] to counteract the cognitive demands of discomfort [72] (Figure 3). Hence, in this regard, it can be hypothesised that improved intestinal health derived from a positive GM profile might positively impact the higher tolerance of sensory limits and effort perceptions in the long term. Therefore, the metabolite dialogue between bacteria and host neural pathways would be crucial in stimulating synaptic plasticity, an essential component of the microbiota–gut–brain axis, and in modifying the neural thresholds of tolerance to fatigue or discomfort [73]. To modify neuronal perception and sensitivity, it is hypothesised that physical and functional changes at the level of individual connections between neurons have occurred [74].

As a premise of the present hypothesis, it has been postulated that the GM profile in each individual may differ depending on the perceptions that evolve from acute to chronic neuropathic injuries, accompanied by pain and impairments in functional performance in athletes [75,76]. Regarding osteoarthritis pain, for example, increases in *Streptococcus* species were associated with knee osteoarthritis [77], while *Coprococcus* species were reduced in individuals with widespread chronic pain [78]. Recently, it was demonstrated that the levels of *Bacteroides* were lower in subjects with reduced tolerance to neuropathic pain. Conversely, the abundance of *Prevotella* was increased in subjects with higher tolerance to pain compared to those with lower levels [64]. Wang et al. [79] reported a reduced abundance of *Bacteroides* and *Faecalibacterium* in subjects with neuropathic pain, but *Escherichia*—*Shigella*, *Lachnoclostridium*, *Blautia*, *Megasphaera*, and *Ruminococcus torques* were elevated. Lower levels of *Faecalibacterium prausnitzii* have been correlated with a higher severity of osteoarthritis in older female adults [80]. Still, higher concentrations have been associated with antinociceptive effects in a rat model of irritable bowel disease [81]. These results, in summary, indicate that the GM composition can change drastically in response to injuries and chronic pain due to the higher inflammatory state in the tissues, which directly increases neural sensitivity [82]. However, whether GM dysbiosis is the consequence of the onset of nociceptive pain has not been elucidated, and it is unknown whether systemic pain, fatigue, and/or negative afferences would be modulated through the gut–brain connection.

## 4. GM-Derived Metabolites: NTs and Neuromodulators and Their Influence on Brain Perceptions

The intensity and duration of endurance exercise suggest that substrate homeostasis changes during the activity and directly impacts central command in the motor cortex, thereby increasing effort perception [62,63]. The perceptions associated with energy homeostasis include the regulation of blood pH and the acid–base equilibrium, which is altered during exercise, leading to increased acidosis. In the intestines, the pH is slightly acidic in the caecum but approaches neutrality in the colon [83]. In the intestines, abnormal pH, promoted by chronic metabolic acidosis due to nutritional habits and intense endurance exercise, may increase the risks of epithelial damage and adverse changes in the GM composition of the colon [84]. The stability of colonic pH appears essential for regulating mucosal integrity, including bicarbonate and lactate production, the bacterial fermentation of carbohydrates, and the mucosal absorption of SCFAs [84]. Therefore, the inefficiency of the intestines in regulating blood pH may negatively affect endurance performance and metabolic efficiency. In this regard, some evidence has reported that certain bacteria (*Veillonella atypica*) may reduce lactate during exercise and increase endurance performance [85,86]. On the contrary, the GM composition that can improve the commensal bacteria to produce SCFAs decreases gut pH, and the transformation of primary to secondary bile acids reduces acidification [87,88].

Signalling molecules derived from the GM include SCFAs, NTs, gut hormones and peptides. The NTs’ influence on neuronal activity is promoted by neuromodulatory molecules, such as dopamine, noradrenaline, serotonin, and GABA, produced by a wide array of bacteria [71]. Additionally, the gut microbiota was shown to play a significant role in the metabolism of tryptophan, which is synthesised mainly in the gut [71]. Considering the important role of the GM in NT and SCFA synthesis, we propose different specific bacteria and genera that have been proposed to produce these molecules (see Table 1).

In this narrative review, we conducted a search of scientific databases, including PubMed and Google Scholar, using the following terms and keywords: gut microbiota, physical exercise, endurance exercise, perceptions, fatigue, pain, discomfort, and nervous system. We included in Table 1 phenotypes related to specific target bacteria and genera that have been described as modulators of different neuromodulators involving NTs and SCFAs. Regarding phenotypes (dopamine, serotonin, acetylcholine, GABA, noradrenaline, emotional balance, butyrate, and propionate), references in such cases describe specific bacteria found to be related to these metabolites. We excluded articles that did not consider the relationship between the GM and the brain axis, as well as its relationship with perceptions and emotions.

### 4.1. The Important Role of SCFAs in Inflammation Regulation and Immune Response for Perceptual Perceptions

SCFAs are produced in the large intestine through the anaerobic fermentation of dietary fibres [9]. These molecules are considered beneficial for gut health and play an important role in maintaining gut barrier homeostasis [201]. SCFAs are produced by bacteria from two main bacterial genera, specifically the *Bacteroidetes* phylum, which mainly produces propionate, and the *Firmicutes* phylum, which produces primarily butyrate [202]. Interestingly, a previous study reported that the concentration of SCFAs changed in line with the alteration of the ratio of *Firmicutes* to *Bacteroidetes* phylum bacteria [203]. In any case, SCFAs play important roles in maintaining gut barrier integrity and modulating gut and systemic inflammation [9]. Gut microbiota dysbiosis has been implicated in altered neurologic pathologies, such as depression, Alzheimer’s, Parkinson’s disease, and autism spectrum disorder, which has been improved with SCFA administration [204]. The abundance of butyrate-producing bacteria, such as those from the *Clostridium, Eubacterium*, and *Butyrivibrio* genera [205], was associated with attenuated pain and tumour necrosis factor-α (TNF-α) levels in a peripheral nerve injury model [206]. In one study of rats exposed to chronic stress, a reduced level of *Bacteroides* bacteria has been identified [207], while bacteria belonging to the *Firmicutes* phylum, such as *Clostridium*, increased, providing benefits for visceral hypersensitivity by inhibiting low-grade inflammation [208].

Therefore, the proportion of *Firmicutes* and *Bacteroidetes* phyla may be associated with pain sensitivity because, indirectly, SCFA production can increase nerve activation in the brain, derived from gut afferences. The SCFA metabolites modulate neurotransmission and increase the expression of enzymes, altering the production of noradrenaline and dopamine [209].

The importance of modulating the immune response is vital to mediate the sensitive response of the neural system. This modulation begins in the gut through the regulation of the GM and its metabolites, such as SCFAs or bile acids, and their conversion from a primary to a secondary pathway.

### 4.2. Neurotransmitters, Gut Peptides, and Vagus Nerve–CNS Communication

Overall, communication between the GM and the CNS largely depends on the vagus nerve (VN), which serves as the central pathway of the parasympathetic nervous system, comprising 80% afferent and 20% efferent fibres. The VN connects vital organs such as the lungs, heart, or intestines with the brain [210] through metabolites and neural discharges, influencing perceptions and behaviour [1,211,212,213,214,215].

In the brain, perceived exertion is expressed through the activity of various regions of the motor cortex [216], which is modulated by changes in the concentration of different NTs such as noradrenaline, dopamine, and serotonin [29,217,218,219]. As can be seen in Figure 3, the interaction between certain nutrients, derived metabolites, and digestive hormones during and after endurance exercise modulates the adaptive regulation of neural perceptions and connections accordingly to individual sensory thresholds of fatigue and discomfort. The individual’s perception of exertion is key to voluntarily and involuntarily continuing through action with different intensities. Therefore, the state of the GM is crucial to regulate NT levels from the gut, including glutamate, dopamine, serotonin, and GABA, and also to produce other important metabolites, such as SCFAs and gut hormones (including orexin, galanin, ghrelin, gastrin, and leptin).

A healthy GM may stimulate positive perceptions through the synthesis of nutrients and metabolites, producing beneficial connections in the CNS. It is possible that chronic interconnection between neural discharges from the gut and neuromodulators of the GM changes perceptions and improves tolerance to pain, exertion, and discomfort. In contrast with this hypothesis, dysbiosis of the GM and local inflammation of the intestines probably promote irregular afferences and sensations. The reduced production of NTs due to lower levels of target bacteria, such as *Lactobacillus* or *Bifidobacterium*, and SCFAs may reduce endurance performance through intolerance to effort and poor perceptions. These concepts are depicted in Figure 3, showing the possible bidirectional pathway between the CNS and GM (afferent and efferent nerves from the hypothalamic/pituitary/adrenal (HPA) axis to the gastrointestinal tract) [220].

Neurotransmitters and gut peptides derived from GM metabolites directly modulate other molecules in the blood, such as catecholamines (dopamine and norepinephrine), serotonin (5-HT), noradrenaline, GABA, acetylcholine, and histamine [71].

Glutamate transfers intestinal sensorial signals to the brain through the VN [221]. It is the predominant excitatory neurotransmitter that activates different areas of the brain, spinal cord, and periphery involved in pain sensation [222]. Specific bacteria which have been reported to produce glutamate are *Lactobacillus plantarum*, *Bacteroides vulgatus*, and *Campylobacter jejuni* [223].

In contrast to glutamate, GABA neurotransmitters modulate sensory neuron activity, producing a net inhibitory effect on nociceptive transmission and decreasing pain and discomfort [224]. Concerning GABA production [225], different species have been reported to be important: *Bifidobacterium* spp., *dentium*, *Bacteroides fragilis*, *Parabacteroides*, *Eubacterium* [226], and some *Lactobacillus* spp. [227].

Other neural signalling has been associated with acetylcholine, which is found in various species, including *Lactobacillus plantarum* [228], *Bacillus acetylcholine* [10], and *B. subtilis*, as well as *Escherichia coli* and *Staphylococcus aureus* [229].

Dopamine plays a key role in motivation, predisposing individuals to continue a task despite feelings of discomfort [230] and increasing its release during 1–2 h of physical activity [231]. Therefore, a reduction in dopamine concentration during exercise may be involved in mechanisms contributing to central fatigue by negatively affecting dopaminergic neurocircuits associated with movement and other areas involved in reward and motivation [232]. Dopamine and noradrenaline are thought to play major, yet opposite, roles in the development of central fatigue [233,234,235,236]. Higher brain dopamine levels, for example, have been described to improve endurance performance [237] through increased arousal, reward, and motivation [217]. These findings coincide with a faster rate of increase in the rating of perceived exertion during exercise [238].

Acworth et al. [239] and Newsholme et al. [240] suggested that elevations in brain 5-HT could also contribute to increased exercise-induced fatigue, as they are associated with decreased sleep quality and mental alertness (although this association is not entirely clear) [240,241]. Thus, the elevation of 5-HT in the brain may evoke significant effects on arousal, lethargy, sleepiness, and mood, likely linked to an altered perception of effort and muscular fatigue [242]. Moreover, it has been demonstrated that the concentration of 5-HT during endurance exercise increases until it reaches its maximal levels at the moment of fatigue or cessation of the activity [242]. Davis and colleagues [230] speculated that elevations of 5-HT during physical activity might contribute to fatigue through the inhibition of dopamine release, suggesting that a low 5-HT/dopamine ratio in the brain favours better performance than a high 5-HT/dopamine ratio [230]. It has been estimated that 90% of the serotonin in humans is produced by intestinal bacteria [102], and the *Corynebacterium* genus is capable of synthesising serotonin [243], which influences mood state, pain perception, and other functions [68]. Similarly, *Enterococcus* influences the production of 5-HT [65], which, together with noradrenaline, modulates the descending modulatory pain of the CNS [17].

### 4.3. The Negative Effects of Gut Dysbiosis on Perceptions and Tolerance to Fatigue

It has been reported that GM modulates cognition, afferences, and mood state through communication with the brain. In this regard, a dysbiosis of the GM may elevate toxins and lipopolysaccharides (LPS) in the blood, activating immune cells and increasing the sensitisation and immune response of the host [244]. Therefore, certain molecules and toxins from the GM can drive mechanical hypersensitivity and mediate pain and fatigue tolerance. Symbionts, commensals, and mutualist bacteria can become pathogenic in situations of dysbiosis, leading to significant tissue inflammation and pathology in the gut, skin, and other barriers, which can result in pain or lower tolerance to discomfort. For example, the fungus *Candida albicans*, an opportunistic pathogen, can maintain a commensal homeostatic relationship with its host; however, under certain conditions, such as stress or dysbiosis, it can become pathogenic and have a net negative effect on its host. The secondary effects of pathogens in the gut include the activation of the immune response and the expression of pathogen recognition receptors (PRRs) by host cells, which can detect pathogen-associated molecular patterns (PAMPs). It has been reported that toxins and PAMPs can act directly on sensory afferences and reduce tolerance to discomfort and pain [244].

A state of chronic gut dysbiosis can lead to increased intestinal permeability. Pathogenic species can produce barrier disruption through direct binding to epithelial cells, such as enteropathogenic *E. coli*, or by secreting toxins, including zonula occludens toxin (ZOT) and hemagglutinin/protease (HA/P) secreted by *Vibrio cholerae*, or the enterotoxin secreted by *Clostridium perfringens* [245]. In summary, a chronic dysbiosis of the GM could change the immune response, with the gut–brain connection involving perceptions and mood states. These conditions may be important in predisposing individuals to tolerate effort during maximal exercise.

## 5. Differences Between Endurance Athletes and the General Population Regarding Gut Microbiota and the Neural System as an Adaptive Response to High Fatigue Exposure

The adaptive biological responses that we are building during our life depend on the most stimuli supported. In this regard, the maximal physiological stimuli supported during exercise by athletes and the general population differ largely. Thus, nutrition strategies associated with exercise in athletes promote changes in their GM in comparison with the general population. The differences between the GM of athletes (even comparing athletic level, elite and sub-elite) and the general population have been demonstrated. In fact, greater GM diversity is related to better endurance performance in elite athletes [246,247,248].

Athletic training and competition coexist with an ample range of sensations and perceptions of fatigue and discomfort. Previous studies have described how fatigue creates specific neural networks and learned feelings associated with physiological limits achieved during exercise [46,249,250]. The maintenance of long periods of tachycardia, hyperventilation, hyperthermia, dehydration with alterations on metabolism homeostasis, and inflammatory processes after exercise change perceptions and thresholds of tolerance on intensity and volume. In these conditions, these perceptions and sensations possibly are not supported by the general population with lower tolerance.

An important goal of athletes is stimulating tolerance to physiological limits and improving fatigue thresholds to reach greater performance in the final and decisional moments of competition. The link between the brain and systemic organs is key to elevate tolerances of discomfort related to heat, hypoxia, metabolic acidosis, or muscular soreness. Although it could be considered not directly related to fatigue, the gut microbiota specialization in athletes possibly provides NTs, SCFAs, and endocrine molecules supporting the brain modulation of neural networks. In fact, the neuronal loop of synapsis related to perceptions and emotions is connected to different molecules such as NTs derived from microbiota [73]. Therefore, a basic threshold for tolerating harder perceptions of discomfort could relate to the specialized GM production of neural metabolites. The specific nature of exercise determines nutrition, absorption, and substrate uptake during and after physical activity. In this regard, providing specific nutrients to feed a favourable GM must be a main goal for athletes but also in the general population. Some studies have postulated a direct correlation of fatigue tolerance in athletes with better physical performance [251,252]. The better effort regulation, self-control, and executive functions such as inhibitory control in athletes due to athletic training increase willpower by changing the connectivity between brain structures [253] through the action of neurotransmitters in the CNS [254].

The hypothesis described here supports the idea that greater dopaminergic activity in athletes in comparison with sedentary subjects can enhance motivation and tolerance to discomfort, increasing performance. Reaching a richer and more diverse GM in athletes in comparison with sedentary subjects would improve the abundance of short-chain fatty acid (SCFA)-producing bacteria contributing to the support of brain activity related to fatigue [255,256].

## 6. Nutritional and Probiotic Interventions to Modulate GM and Change Effort Sensitivity: The Hypotheses

Precision nutrition approaches are being developed to provide comprehensive and dynamic nutritional recommendations tailored to individual variables, including microbiota, health status, and dietary patterns [257]. Generally, microbiota-based precision nutrition approaches seek to optimise GM and its modulating ability. In this regard, probiotics-based precision nutrition has been preliminarily explored to modulate the GM–brain axis using mainly probiotics [258], which are defined as live microbes that, when introduced in adequate quantities, confer health benefits on the host [259]. Furthermore, according to the influence of probiotics reported in other areas, such as against several GM dysbiosis conditions, or the positive effect, even in clinical trials, on chemotherapy side effects [260] and infections [261], we could hypothesise their positive results in sports, performance, and effort tolerance-related areas in the future. However, despite promising results, the consumption of probiotics in food and/or supplements has shown a common limitation in bioavailability and effect duration, regardless of the area of application, as they are not maintained in the intestine [262].

To address this limitation, microbiome-based precision nutrition is recommended, including probiotics and prebiotics to promote the growth of beneficial gut microbes, thereby favouring the natural development of the GM and potentially extending its metabolic effects [263]. In other words, combining probiotics and prebiotics produces the first bioactive metabolites directly in the human gut. Secondly, stimulating target bacteria not only to colonise but also to maintain themselves in the intestine, consequently, has a potential long-term effect.

Although the International Society of Sports Nutrition (ISSN) concluded that probiotics have strain-specific effects in athletes [264], most studies associate effort tolerance-related phenotypes with specific bacteria at the genus or species level (see Table 1). Therefore, selecting target bacteria associated with effort tolerance-related phenotypes should be done by prudent criteria and should be investigated in the future.

In summary, current evidence allows for the description of target bacteria with some of the most functionally evidenced effort tolerance-related phenotypes associated with neurotransmitters and gut metabolites. In this sense, several studies have associated different bacteria with neurotransmitter regulation, specifically regarding dopamine [71,89,90,91,92,93,94,95,96,97,98,99,100], serotonin [71,89,90,91,92,93,94,101,102,103,104], acetylcholine [71,89,91,92,105,106,107,108,109,110], GABA [71,91,92,93,111,112,113,114,115,116,117,118,119], and noradrenalin [71,91,92,93,98,120] (see Table 1). Similarly, there is increasing evidence of the contribution of specific bacteria to emotional balance [98,103,114,121,122,123,124,125,126,127,128,129,130,131,132,133,134,135,136,137,138,139]. The GM influence on its phenotype appears to be related, at least in part, to neurotransmitter regulation due to the reported bacteria’s psychoactive ability; therefore, they have been preliminarily selected as target bacteria. On the other hand, as previously reported, the SCFAs’ pivotal role, specifically butyrate [140,141,142,143,144,145,146,147,148,149,150,151,152,153,154,155,156,157,158,159,160,161,162,163,164,165,166,167,168,169,170,171,172,173,174] and propionate [147,158,160,175,176,177,178,179,180,181,182,183,184,185,186,187,188,189,190,191,192,193,194,195,196,197,198,199,200] (see Table 1), in the overall functionality of the GM, and potentially in effort tolerance, is strongly supported by the scientific literature.

It is worth noting that part of the current scientific evidence on a bacterium’s ability to produce a specific metabolite is based on in vitro studies. In vitro studies are essential for generating knowledge, but they are insufficient for determining whether a bacterium behaves the same way in a real biological system. To assess the potential of a bacterium on human health, in vivo studies must be performed. For example, to determine if a bacterium meets the requirements of a potential probiotic strain, various in vitro tests are first conducted. Following this, in vitro and in vivo safety assessments, clinical trials, and in vivo efficacy assessments are conducted to ensure the product’s safety and effectiveness in humans. There is no universal international standard to determine the safety of a probiotic for humans. Suppose a bacterium is considered safe for humans. In that case, it is classified as “Generally Recognised as Safe” (GRAS) by the United States Food and Drug Administration, or it is included in the “Qualified Presumption of Safety (QPS) list” by the European Food Safety Authority (EFSA) [265].

The bioavailability of GM key metabolites, neurotransmitters, and neuromodulators is affected by the cooperative and competitive relationships between bacteria within the microbial community. Cooperative relationships primarily refer to cross-feeding, where a product generated by one bacterium is utilised as a substrate by another [266]. There is an example of bidirectional cross-feeding between target bacteria *Akkermansia muciniphila* and *Eubacterium hallii*. *A. muciniphila* degrades host mucus and generates 1,2-propanediol that supports the growth of *E. hallii*. In return, *E. halli* produces pseudovitamin B12, which is used by *A. muciniphila* as a cofactor for converting succinate to propionate [151]. Bacteria compete to utilise the same energy resource, as observed for target bacteria *A. muciniphila* and *Bacteroides thetaiotaomicron*, which are both mucolytic [151,267].

The final target-bacteria selection is detailed in Table 1, indicating which bacteria have been associated with each effort tolerance-related phenotype and the corresponding literature references in each case. As can be seen, there is currently abundant and coherent scientific evidence regarding effort tolerance-related phenotypes and GM. Interestingly, some target bacteria have been associated with regulating more than one phenotype, such as *Streptococcus thermophilus* and *Lactobacillus plantarum*, both of which are involved in regulating acetylcholine, dopamine, and serotonin, and the latter is also associated with regulating GABA (Table 1). In addition, it is noteworthy that 17 of the 65 target bacteria listed in Table 1 are considered probiotics and are included in the revised list of microorganisms, with QPS-recommended microorganisms, for safety risk assessments carried out by the EFSA (https://www.efsa.europa.eu/en/topics/topic/qualified-presumption-safety-qps, accessed on 26 July 2025). The target bacteria included in the EFSA list are *Bifidobacterium* (*B. adolescentis*, *B. animalis*, *B. bifidum*, *B. breve*, and *B. longum*), *Lactobacillus* (*L. acidophilus*, *L. brevis*, *L. casei*, *L. delbrueckii*, *L. helveticus*, *L. paracasei*, *L. plantarum*, *L. reuteri*, *L. rhamnosus*, *L. salivarius*, and *L. lactis*), and *Streptococcus thermophilus*.

Therefore, we propose that these types of target bacteria have a high potential to influence effort tolerance and overall health positively, and are especially promising as targets in microbiome-based precision nutrition. Hence, a microbiome-based precision nutrition approach is proposed for effort tolerance-related GM optimisation (comprising phenotypes related to the regulation of the neurotransmitters dopamine, serotonin, acetylcholine, GABA, and noradrenaline, as well as emotional balance and the SCFAs butyrate and propionate).

## 7. Limitations

Despite the strong evidence presented here regarding the potential of microbiome-based precision nutrition, the practical application of this food recommendation should be tested in future longitudinal studies, where different individuals are analysed before and after a precision nutrition intervention to improve effort tolerance by GM in the mid- to long-term. Although it is possible to change the composition of the microbiota through food in the short term, the objective should be to maintain these changes in microbiota composition [268] and, consequently, that the potential benefit in effort tolerance would be durable. In this regard, a microbiome-based precision nutrition approach could facilitate the individual identification of target diet components by a nutrition professional, who must consider the specific requirements of each person due to food allergies or incompatibilities between specific diet components and particular medications or treatments. The present investigation of this topic is scarce, and no previous studies have proposed a similar hypothesis as described here. For this reason, some of the proposals presented here cannot be demonstrated or replicated. Although we inform the individual of their fatigue and discomfort threshold during maximal exercise, we do not present any proposal for measurement. The Borg scale has been employed for decades, connecting individual perceptions to physiological limits during maximal effort competence. Therefore, a separate analysis of the GM and different NTs related to maximal effort tolerance could be proposed for future study. These limitations observed here are the same as those observed in other studies which recently describe the connection between some neuropsychological pathologies, such as depression, anxiety, hyperactivity, and even neurodegenerative diseases. Despite the present limitations, this article could contribute to considering the GM state in states of physical underperformance associated with fatigue or discomfort tolerance.

## 8. Conclusions

An appropriate management of tolerance to effort and discomfort during intense endurance exercise is a key factor for successful performance in competition. Higher tolerance for effort and discomfort is certainly an advantage for competitive athletes. In this context, the GM state is physiologically important for the athlete’s health, exercise performance, and competition success, which is also associated with improved sensory effort tolerance. Previous studies in other medical areas have suggested that the GM composition can improve pain tolerance during degenerative injuries, such as osteoarthritis, as well as in medical treatments, including chemotherapy or post-surgery states. Concerning endurance performance, only a few data are available on the potential role that certain GM phenotypes could play in increasing the maximal tolerance to effort and fatigue. Here, we hypothesise that GM status may modify cognitive processes of discomfort, pain, and/or effort during intense exercise through the modulation of systemic and local inflammation, gut barrier permeability, vagal signalling, and the production of metabolites and neurotransmitters in the intestines, as well as their direct connection with the brain. Furthermore, we propose specific bacteria that have been described as modulators of different GM metabolites related to the regulation of pain, discomfort, and effort tolerance through the modulation of gut peptides, hormones, and neurotransmitters. The GM is modified through exercise in athletes, but at the same time precision nutrition and probiotics may be optimal for modulating specific perceptual responses and brain sensations. Future studies should evaluate whether perceptions resulting from high-intensity training are necessary to stimulate the GM and the increase in particular bacteria related to the favourable modulation of gut permeability and anti-inflammatory levels. It is reasonable to think that a favourable GM state could improve CNS connections related to effort tolerance, stimulating areas of the brain involved in emotions and perceptions.

## Figures and Tables

**Figure 1 nutrients-17-02836-f001:**
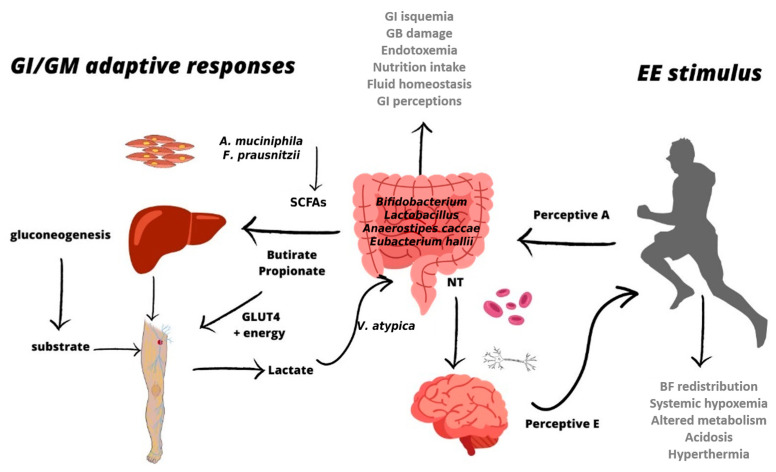
Endurance exercise (EE) is the essential stimulus for promoting effort tolerance in athletes and GM phenotypic adaptive changes. EE promotes gastrointestinal (GI) ischemia, gastrointestinal damage, and intestinal permeability, which are aggravated by inadequate nutrition and hydration during EE. EE stimulates the redistribution of blood flow (BF) to the active muscles, promoting hyperthermia and systemic hypoxemia. The gut microbiota (GM) plays a crucial role in producing neurotransmitters (NTs) in the gut, primarily through bacteria such as *Bifidobacterium, Lactobacillus*,* Anaerostipes caccae*, and *Eubacterium hallii*. Thus, some bacteria, such as *Akkermansia muciniphila* and *Faecalibacterium prausnitzii*, specifically produce SCFAs. NTs produced in the gut may contribute to altering physiological perceptions directly through neural signals from the gut to the brain, and perceptual thresholds (perceptual E) related to EE are supported during daily activities. EE would be a necessary stimulus to increase the NT threshold production from the gut, thereby modulating exertion tolerance.

**Figure 2 nutrients-17-02836-f002:**
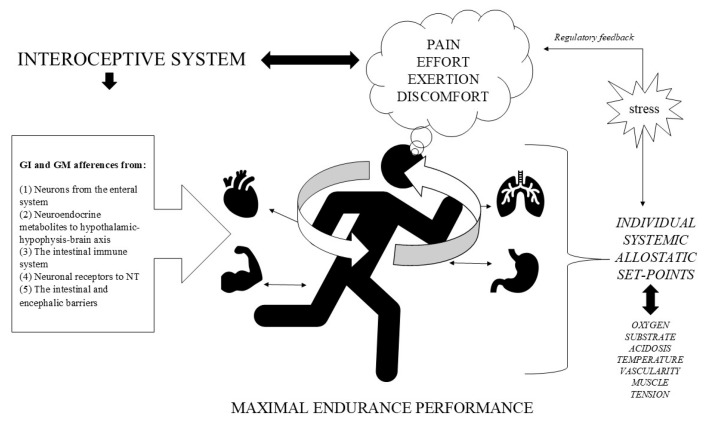
Relationship between maximal endurance performance, the interoceptive system, and regulatory feedback. During maximal effort, athletes feel sensations of pain and discomfort which are derived from different peripheral systems and processed by the brain. Everyone has a different set point of tolerance for stress, pain, and exertion, according to their systemic allostatic thresholds. The interoceptive system has the important function of regulating perceptions from afferent inputs, including specialised cells, organs, and active tissues such as muscles. The gut and microbiota also produce afferences from the neurons, metabolites, and enterocytes during endurance exercise and recovery processes.

**Figure 3 nutrients-17-02836-f003:**
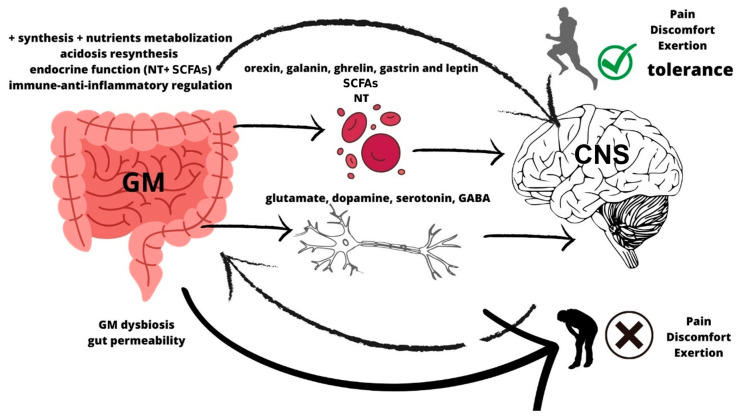
Hypothesis from interactions between GM and CNS afferences related to perceptive tolerance. The GM acts as a key neuroendocrine organ, directly connected to the CNS through neural afferences, NT production, and hormone levels.

**Table 1 nutrients-17-02836-t001:** List of target bacteria associated with different phenotypes related to effort tolerance (supra-phenotype): regulation of neurotransmitters, dopamine, serotonin, acetylcholine, GABA, and noradrenalin; regulation of emotional balance and SCFA butyrate and propionate. Filled gaps indicate an association reported between a target bacterium and a specific phenotype. Blank gaps indicate the absence of association.

Supraphenotype	Effort Tolerance
Phenotypes	Dopamine	Serotonin	Acetylcholine	GABA	Noradrenalin	Emotional Balance	Butyrate	Propionate
**Target-Bacteria**		[71,89,90,91,92,93,94,95,96,97,98,99,100]	[71,89,90,91,92,93,94,101,102,103,104]	[71,89,91,92,105,106,107,108,109,110]	[71,91,92,93,111,112,113,114,115,116,117,118,119]	[71,91,92,93,98,120]	[98,103,114,121,122,123,124,125,126,127,128,129,130,131,132,133,134,135,136,137,138,139]	[140,141,142,143,144,145,146,147,148,149,150,151,152,153,154,155,156,157,158,159,160,161,162,163,164,165,166,167,168,169,170,171,172,173,174]	[147,158,160,175,176,177,178,179,180,181,182,183,184,185,186,187,188,189,190,191,192,193,194,195,196,197,198,199,200]
	**References**
*Akkermansia muciniphila*								
*Alistipes putredinis*								
*Anaerostipes*								
*Anaerostipes caccae*								
*Anaerostipes hadrus*								
*Anaerotruncus colihominis*								
*Bacillus*								
*Bacteroides*								
*Bacteroides thetaiotaomicron*								
*Bacteroides uniformis*								
*Bacteroides vulgatus*								
*Bifidobacterium*								
*Bifidobacterium adolescentis*								
*Bifidobacterium animalis*								
*Bifidobacterium bifidum*								
*Bifidobacterium breve*								
*Bifidobacterium dentium*								
*Bifidobacterium longum*								
*Blautia*								
*Blautia coccoides*								
*Blautia obeum*								
*Butyricimonas*								
*Butyrivibrio*								
*Clostridium butyricum*								
*Clostridium leptum*								
*Collinsella aerofaciens*								
*Coprococcus*								
*Coprococcus catus*								
*Coprococcus eutactus*								
*Eubacterium*								
*Eubacterium hallii*								
*Eubacterium limosum*								
*Eubacterium rectale*								
*Faecalibacterium*								
*Lachnospira*								
*Lactobacillus*								
*Lactobacillus acidophilus*								
*Lactobacillus brevis*								
*Lactobacillus casei*								
*Lactobacillus delbrueckii*								
*Lactobacillus helveticus*								
*Lactobacillus paracasei*								
*Lactobacillus plantarum*								
*Lactobacillus reuteri*								
*Lactobacillus rhamnosus*								
*Lactobacillus salivarius*								
*Lactococcus*								
*Lactococcus lactis*								
*Megasphaera*								
*Megasphaera elsdenii*								
*Odoribacter*								
*Oscillibacter*								
*Oscillospira*								
*Parabacteroides*								
*Phascolarctobacterium*								
*Prevotella*								
*Propionibacterium freudenreichii*								
*Roseburia*								
*Roseburia faecis*								
*Roseburia hominis*								
*Roseburia intestinalis*								
*Roseburia inulinivorans*								
*Streptococcus thermophilus*								
*Subdoligranulum*								
*Veillonella*

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
