# Peer review of "A Narrative Hypothesis: The Important Role of Gut Microbiota in the Modulation of Effort Tolerance in Endurance Athletes"

_nutrients, 2025, doi:10.3390/nu17172836_

Round 1
Reviewer 1 Report
Comments and Suggestions for Authors
The study examines the link between gut microbiota and regulation of the perception of fatigue and pain during maximal resistance exercise. The so-called "microbiota-brain axis" can affect emotional stability and physical well-being, helping athletes to better tolerate exertion and discomfort. Through a review of the literature, the authors highlight that exercise and nutrition can positively modify the microbiota, increasing tolerance to fatigue. In conclusion, the microbiota seems to play a decisive role in improving endurance performance, raising the threshold of perception of fatigue and pain.
The study is interesting, simple, and straightforward to understand.
The Abstract provides an exhaustive overview of what will be covered in the following sections.
Write "Objectives" in bold.
Introduction: this section is well developed, however there are improvements to be made:
Line 81: Please revise the expression “gut bowel damage”, as it is redundant and uncommon in scientific English. Consider using “intestinal damage”, “bowel injury”, or “gastrointestinal injury” instead.
Line 87: The definition of SCFAs is repeated in full here, but it was already provided at line 52. Please avoid redundancy.
At line 85, neurotransmitters were defined as NTs, but this abbreviation was not used consistently at line 96. For consistency, the term at line 96 should also appear as NTs.
Paragraph 2: this section is well developed.
Paragraph 3: this section is well developed, however there are improvements to be made:
Line 195: NTs was already defined earlier in line 85.
Line 221: Since this is the first occurrence of GABA, it is recommended to provide the full term (gamma‑aminobutyric acid) before using the abbreviation.
Line 222: The definition of SCFAs is repeated in full here, but it was already provided at line 52. Please avoid redundancy.
Line 226: The definition of NTs is repeated in full here, but it was already provided at line 85. Please avoid redundancy.
Paragraph 4: this section is well developed, however there are improvements to be made:
Line 274: The definition of SCFAs is repeated in full here, but it was already provided at line 52. Please avoid redundancy.
Line 320-323: The concept in this sentence is currently explained through the description of a previous figure. It would be clearer to present the concept directly in the text and then simply refer to the figure to provide visual support.
Paragraph 5: this section is well developed.
The Limitations of the study are clearly stated, which is appreciated; however, further elaboration or discussion on some aspects could strengthen the manuscript.
The combined Discussion and Conclusion section works well and is clearly presented; however, expanding the argument slightly could improve clarity and provide more depth to the manuscript.
The References are relevant; however, we kindly suggest reviewing them to ensure that all are formatted consistently and follow the same citation style.
The document is written in good English.
In summary, the article is well-structured and addresses a relevant topic, but it can be improved with simple adjustments.
Author Response
Thank you very much for your work. We consider that your contribution improve substantially the present manuscript.
The Abstract provides an exhaustive overview of what will be covered in the following sections.
Write "Objectives" in bold.
Thanks. Done.
Introduction: this section is well developed, however there are improvements to be made:
Line 81: Please revise the expression “gut bowel damage”, as it is redundant and uncommon in scientific English. Consider using “intestinal damage”, “bowel injury”, or “gastrointestinal injury” instead.
Response 1: Right. We have changed for gastrointestinal permeability because is more complete and include damage both for gastric and intestinal tissues.
Line 87: The definition of SCFAs is repeated in full here, but it was already provided at line 52. Please avoid redundancy.
Response 2: Thank you. Ths phrase has been reformulated to only include SCFAs and productor bacteria such as Akkermansia and Faecalibacterium.
At line 85, neurotransmitters were defined as NTs, but this abbreviation was not used consistently at line 96. For consistency, the term at line 96 should also appear as NTs.
Response 3: You are right. Changed!
Line 195: NTs was already defined earlier in line 85.
Response 4: Thank you. Done.
Line 221: Since this is the first occurrence of GABA, it is recommended to provide the full term (gamma‑aminobutyric acid) before using the abbreviation.
Response 5: Correct. Full term has been included in the text.
Line 222: The definition of SCFAs is repeated in full here, but it was already provided at line 52. Please avoid redundancy.
Response 6: Done.
Line 226: The definition of NTs is repeated in full here, but it was already provided at line 85. Please avoid redundancy.
Response 7: Done.
Line 274: The definition of SCFAs is repeated in full here, but it was already provided at line 52. Please avoid redundancy.
Response 8: Done and also the NTs term which was also with full term.
Line 320-323: The concept in this sentence is currently explained through the description of a previous figure. It would be clearer to present the concept directly in the text and then simply refer to the figure to provide visual support.
Response 9: The description of Figure 3 has been completely removed and included in the text (lines 312-328).
The Limitations of the study are clearly stated, which is appreciated; however, further elaboration or discussion on some aspects could strengthen the manuscript.
Response 10: Thank you very much for your suggestion. We have try to improve some aspects of this section including other limitations in different medical areas. See lines 492-504.
The combined Discussion and Conclusion section works well and is clearly presented; however, expanding the argument slightly could improve clarity and provide more depth to the manuscript.
Response 11:
The References are relevant; however, we kindly suggest reviewing them to ensure that all are formatted consistently and follow the same citation style.
Response 12: Ok. We are revising.
Other changes has been performed accordingly other reviewers.

Reviewer 2 Report
Comments and Suggestions for Authors
The authors have carried out a review of the importance of gut microbiota for endurance exercise and effort tolerance in athletes. The review is well written and well illustrated. Table 1 is particularly well presented.
Unfortunately, there is no evidence that this review was conducted systematically. There is just one methodology sentence (in the abstract). There is no indication about how papers were chosen to be included in this review. The dates of the database searches are not given. What were the inclusion and exclusion criteria? Overall, it would not be possible for other researchers to replicate this review.
Author Response
Thank you very much for your revision and suggestion to improve the present manuscript.
The present review is not a systematic or metaanalysis review as you may have suggested. Here we are proposing a new perspective on perceptive sensations during endurance exercise and their association with the gut microbiota-brain axis. This new topic has been not previosuly discussed and it is impossible include other reproductive articles here. However we are improved and changed a lot the present manuscript accordingly to precise recommendation of reviewers. We have change the tittle for example to narrative hypothesis. You can see changes in the attached document.
We are very grateful with you if precisely propose arguments to improve the manuscript with the narrative review perspective.
The main premise here described is based on how microbiota-brain axis modulate perceptions and regular mood state.
Please see the attachment
Thanks

Reviewer 3 Report
Comments and Suggestions for Authors
Review Report:
Titel: Evaluation of “A Hypothesis: An Important Role of Gut Microbiota in the Modulation of Effort Tolerance in Endurance Athletes”
General Assessment
This manuscript addresses a highly relevant and emerging topic, hypothesizing that the gut microbiota plays a critical role in modulating effort tolerance, discomfort, and pain perception in endurance athletes through the gut–brain axis. The review is comprehensive, covering physiological, neurological, and microbiological aspects, and integrates an extensive number of references. However, the current version is predominantly theoretical and presents several methodological and structural weaknesses that limit its scientific impact. Major revisions are required to improve clarity, methodological rigor, and the strength of evidence supporting the conclusions.
Weaknesses
- The manuscript is primarily hypothesis-driven with insufficient empirical support and lacks any clearly defined sample size or experimental methodology.
- Participant selection criteria, inclusion/exclusion parameters, and data collection procedures are insufficiently described, reducing reproducibility.
- Potential confounding variables are not considered, which weakens the validity of the interpretations.
- Conclusions are often overinterpreted, suggesting causal effects despite being based on correlational or theoretical data.
- Alternative explanations for observed or cited associations are not critically discussed.
- Figures and tables lack sufficient labeling and detailed legends for standalone interpretation.
- Key findings are reported without confidence intervals or effect sizes, limiting their scientific rigor.
Author Response
- The manuscript is primarily hypothesis-driven with insufficient empirical support and lacks any clearly defined sample size or experimental methodology.
- Participant selection criteria, inclusion/exclusion parameters, and data collection procedures are insufficiently described, reducing reproducibility.
- Potential confounding variables are not considered, which weakens the validity of the interpretations.
- Conclusions are often overinterpreted, suggesting causal effects despite being based on correlational or theoretical data.
- Alternative explanations for observed or cited associations are not critically discussed.
- Figures and tables lack sufficient labeling and detailed legends for standalone interpretation.
- Key findings are reported without confidence intervals or effect sizes, limiting their scientific rigor.
Dear reviewer,
Thank you very much for your work on our manuscript.
We believe that your contributions are very relevant to improving the text.
We have made substantial changes to the manuscript in accordance with your suggestions and those of other reviewers.
With regard to the methodology and inclusion and exclusion criteria, you are correct. It is not a critical review but rather a narrative one. This has been changed in the title and abstract, and it has been described at various points throughout the manuscript.
The changes included in the article can be seen in red for subsequent acceptance.
Regarding point 1, there are no previous studies that have described this hypothesis in endurance exercise. We have included previous studies that have hypothesized about the state of the gut microbiota and pain or perceptions associated with mood. Therefore, it is not a study with a sample size.
This argument connects to point 2. No sample has been analyzed so far, and it is a hypothesis that can be reinforced by the arguments described throughout the manuscript.
With regard to other variables that may influence and have not been described, this is true; however, we focus on the relationship between the intestinal ecosystem and its relationship with the central nervous system and immunity. The perception of effort, pain, or discomfort is subjective, and the variables described summarize in part what alters sensations: altered metabolism, hypoxia, temperature, etc. The present narrative review desire to describe how altered gut health and microbiota may signaling chronically to CNS changing neural discharges and levels of neuromodulators involved in sensations (Neurotransmitters and SCFAs).
Please, review the new version and provide further recommendations.
Please see the attachment R3

Round 2
Reviewer 1 Report
Comments and Suggestions for Authors
I would accept the paper nin this form
Author Response
Thank you very much for your work and contribution to improve the present manuscript.
Reviewer 2 Report
Comments and Suggestions for Authors
I thank the authors for the changes they have made. In particular, I am pleased that the title makes it clear that this submission is a narrative hypothesis. Also, the introduction is clearer in respect of the motivation of this narrative review. The limitations section is much improved.
Author Response
Thank you very much for your contribution with the present manuscript.
Your suggestion have improved substantially the article.
Reviewer 3 Report
Comments and Suggestions for Authors
None
Author Response
Thank you very much for your contribution to improve the present manuscript.